# Confirmation of Large Language Models in Head and Neck Cancer Staging

**DOI:** 10.3390/diagnostics15182375

**Published:** 2025-09-18

**Authors:** Mehmet Kayaalp, Hatice Bölek, Hatime Arzu Yaşar

**Affiliations:** 1Department of Medical Oncology, Ankara University Faculty of Medicine, Ankara University, Ankara 06620, Turkey; kayaalpmehmet2728@gmail.com (M.K.); hati.kocc@gmail.com (H.B.); 2Ankara University Cancer Research Institute, Ankara University, Ankara 06620, Turkey

**Keywords:** artificial intelligence, large language models, head neck cancers

## Abstract

**Background/Objectives**: Head and neck cancer (HNC) is a heterogeneous group of malignancies in which staging plays a critical role in guiding treatment and prognosis. Large language models (LLMs) such as ChatGPT, DeepSeek, and Grok have emerged as potential tools in oncology, yet their clinical applicability in staging remains unclear. This study aimed to evaluate the accuracy and concordance of LLMs compared to clinician-assigned staging in patients with HNC. **Methods**: The medical records of 202 patients with HNC, who presented to our center between 1 January 2010 and 13 February 2025, were retrospectively reviewed. The information obtained from the hospital information system by a junior researcher was re-evaluated by a senior researcher, and standard staging was completed. Except for the stage itself, the data used for staging were provided to a blinded third researcher, who then entered them into the ChatGPT, DeepSeek, and Grok applications with a staging command. After all staging processes were completed, the data were compiled, and clinician-assigned stages were compared with those generated by the LLMs. **Results**: The majority of the patients had laryngeal (45.5%) and nasopharyngeal cancer (21.3%). Definitive surgery was performed in 39.6% of the patients. Stage 4 was the most common stage among the patients (54%). The overall concordance rates, Cohen’s kappa values, and F1 scores were 85.6%, 0.797, and 0.84 for ChatGPT; 67.3%, 0.522, and 0.65 for DeepSeek; and 75.2%, 0.614, and 0.72 for Grok, respectively, with no statistically significant differences between models. Pathological and surgical staging were found to be similar in terms of concordance. The concordance of assessments utilizing only imaging, only pathology notes, only physical examination notes, and comprehensive information was evaluated, revealing no significant differences. **Conclusions**: Large language models (LLMs) demonstrate relatively high accuracy in staging HNC. With careful implementation and with the consideration of prospective studies, these models have the potential to become valuable tools in oncology practice.

## 1. Introduction

Head and neck cancers (HNCs) represent a significant global health issue, with approximately 946,456 new cases and 481,001 cancer-related deaths reported in the 2022 GLOBOCAN data [1]. The most common subtypes of HNCs include cancers of the lip and oral cavity, larynx, nasopharynx, oropharynx, and hypopharynx. The etiology of head and neck cancers includes environmental factors such as tobacco and alcohol consumption, viral agents including human papillomavirus (HPV) and Epstein–Barr virus (EBV), and genetic syndromes. The treatment of head and neck cancers includes surgical resection, definitive chemoradiotherapy, and trimodal therapy, with the TNM staging system serving as a key prognostic factor in determining the optimal therapeutic approach [2].

The oral cavity is bounded anteriorly by the lip mucosa and posteriorly by the anterior tonsillar region. The nasopharynx extends from the clivus superiorly to the hard and soft palate inferiorly. The oropharynx ranges from the circumvallate papillae to the posterior pharyngeal wall. The hypopharynx includes the piriform sinuses and the postcricoid area. The larynx extends from the supraglottic region to below the cricoid cartilage. The paranasal sinuses include the maxillary, ethmoidal, sphenoid, and frontal regions. The salivary glands consist of the parotid, submandibular, and sublingual glands.

In the staging algorithm, following a physical examination and endoscopic assessment conducted by an otolaryngologist specialized in oncology, imaging modalities such as computed tomography (CT), magnetic resonance imaging (MRI), and positron emission tomography (PET), along with histopathological evaluation via biopsy, are essential for comprehensive assessment [3]. The most common histological variant is squamous cell carcinoma; however, adenocarcinoma in glandular tumors, as well as melanoma and sarcoma, may also be observed. Testing for EBV and HPV, which are potential etiological factors for squamous cell carcinoma, is essential for staging, prognosis, and follow-up.

In the staging of head and neck cancer, the American Joint Committee on Cancer (AJCC) TNM 8 staging system is utilized, which may vary based on clinical, pathological, and molecular characteristics [4]. Preoperative clinical staging is performed not only based on tumor size but also according to anatomical landmarks. The necessity of evaluation by a specialized radiologist and the omission of critical anatomical landmarks in imaging reports complicate the staging process. Separate staging systems are utilized for laryngeal cancer, including the supraglottic, glottic, and subglottic regions, as well as for sinus cancers, which encompass the maxillary sinus, ethmoid sinus, and nasal cavity. In the staging of these malignancies, along with nasopharyngeal cancer, anatomical localization is prioritized over tumor size. Additionally, oropharyngeal cancer staging incorporates two distinct systems based on p16 positivity. Given that both clinical and pathological staging are applied to all head and neck cancers except nasopharyngeal cancer, the staging process is considered complex. These factors raise an important question about the feasibility of employing artificial intelligence applications as tools in the staging process.

Large language models (LLMs) have recently transformed the landscape of artificial intelligence, demonstrating exceptional capabilities in natural language processing (NLP) tasks [5]. These advances stem from the evolution of neural architectures, particularly transformers, alongside increased computational power and expansive training datasets. LLMs can perform complex language-related tasks such as translation, summarization, information retrieval, and conversational interactions with human-like proficiency. Natural language processing (NLP) models, through their integration with fields such as proteomics, are increasingly contributing to oncology by supporting tasks like tumor T-cell antigen (TTCA) prediction, which is critical for the development of personalized cancer therapies [6,7]. The scalability of these models—ranging from billions to hundreds of billions of parameters—has significantly improved their performance and generalization capabilities.

The evolution of large language models began in 2019 with T5 and mT5. In 2020, OpenAI introduced GPT-3, marking a major advancement in AI. In 2021, models like Codex, Jurassic-1, LaMDA, and GLM were developed. By 2022, LLaMA, BLOOM, PaLM, and OPT gained prominence, while 2023 saw the rise of LLaMA 2, Falcon, Mistral, and StarCoder. By 2024, models like GPT-4o, Gemini 1.5, Claude, Mixtral 8x22B, and DeepSeek V2 represented the latest advancements in artificial intelligence [5]. Launched in January 2025, the China-based DeepSeek-R1, developed with cost-efficient methods and limited computational resources, garnered significant attention for exhibiting performance comparable to that of ChatGPT o1 [8]. Grok is an LLM-based chatbot developed by xAI, with access to X (formerly Twitter).

Numerous studies have been conducted to validate LLM models, particularly in the fields of oncology and medicine. Although these models aim to improve healthcare accessibility, they may yield inaccurate outcomes that can result in medical errors [9].

Most oncology-related research focuses on confirming established medical knowledge, while studies evaluating patient data remain relatively scarce [10].

As pre-trained language models have been tested in cancer staging [11], studies on the use of widely accessible and commonly used LLM models for staging are rapidly emerging in the literature [11]. Staging studies conducted with ChatGPT, which is considered the most commonly used chatbot in head and neck cancers in general and oropharyngeal cancers in particular, are available [12,13]. However, no such study has yet been conducted for DeepSeek or Grok.

This study was designed to explore the use of ChatGPT while also evaluating DeepSeek and Grok for the first time in both oncology and head and neck cancer staging, aiming to assess their potential in clinical applications.

## 2. Materials and Methods

### 2.1. Patient Group

A total of 202 patients with head and neck cancer who presented to Ankara University medical oncology clinic between 1 January 2010 and 13 February 2025 were included in the study. Patient files were compiled by reviewing outpatient clinic notes to obtain clinical examination records (physical examination, endoscopic images), radiological imaging [ultrasonography (USG), CT, MRI, PET-CT, etc.], pathology results, and surgical notes. Patients with recurrence were excluded due to possible anatomical alterations following treatments.

The sample size of 202 patients in our study provides approximately 85% power to detect a 10% absolute difference in diagnostic accuracy (e.g., 76% vs. 86%) at a two-sided significance level of 0.05, based on estimates derived from the study by Tozuka et al., which used 100 fictional cases to evaluate large language model performance in lung cancer staging [14].

### 2.2. Staging by Clinicians

The compiled clinical notes were initially staged by a junior medical oncologist, and subsequently reviewed by a senior medical oncologist to establish a reference standard staging. Staging was performed according to the AJCC 8th edition TNM classification. Pathological staging was used for patients who underwent definitive surgery, whereas clinical staging was applied for patients who did not undergo definitive surgery.

### 2.3. LLM-Based Cancer Staging

After clinician-assigned staging was entered into the database, the patient files containing the information required for staging were forwarded to a third researcher who was blinded to the final staging results. This researcher submitted the anonymized data to ChatGPT, Grok, and DeepSeek, using only the prompt “What is the TNM stage?”. Appendix A illustrates an example of this process. To better simulate a real-world clinical scenario and to evaluate both accurate and hallucinated outputs, no prompt engineering was applied. Instead, a single standardized query was used across all cases, reflecting the way a clinician might naturally interact with such systems rather than optimizing prompts for maximal performance. Due to the absence of analysis limitations, ChatGPT Plus 4O [15] was used for ChatGPT, V3 for DeepSeek [16], and Grok3 for Grok [17]. All models were accessed via publicly available open web interfaces. Patient data intended for uploading to LLMs were anonymized by removing personal identifiers such as name, age, city, hospital information, and pathology numbers. Each patient was analyzed in a separate chat session to prevent potential self-training effects within a session.

### 2.4. Statistical Analysis

The staging results were entered into SPSS Version 26 [18], where the clinician-assessed staging data were also recorded, and the analysis was conducted. The chi-square test was used to assess the relationship between categorical variables, while Fisher’s exact test was applied for small sample groups. Cochran’s Q test was used to analyze three or more dependent categorical variables. Following a significant Cochran’s Q result, McNemar tests were performed for pairwise post hoc comparisons between models. Categorical agreement between large language models (LLMs) and the reference staging was assessed using Cohen’s kappa coefficient, which quantifies inter-rater agreement beyond chance. Kappa values were interpreted based on Landis and Koch’s criteria. For performance evaluation, macro-averaged F1 scores were calculated for each model by comparing their predicted T, N, M, and overall stage values with the corresponding ground truth labels on a case-by-case basis. All *p*-values were calculated based on a 95% confidence level. For all pairwise comparisons involving either McNemar tests or kappa coefficient comparisons, statistical significance was determined using the Bonferroni correction.

### 2.5. Ethical Approval and Informed Consent

Ethical approval was obtained from the Ankara University Faculty of Medicine Human Research Ethics Committee on 13 March 2025, with the approval number 2025/168. All researchers adhered to the principles of the Declaration of Helsinki and provided their signed consent. Given the retrospective design and use of fully anonymized data, the requirement for individual informed consent was waived by the Ethics Committee.

The workflow of the study is shown in Figure 1.

## 3. Results

### 3.1. Demographics and Patient Characteristics

The majority of the patients were male (74.8%). Regarding smoking history, 37.1% were active smokers, 36.1% were former smokers, and 25.2% had never smoked. The median age was 58.35 (SD: 14.35), with the most common disease localization being the larynx (45.5%). Only 39.6% of the patients underwent definitive surgery. Patient characteristics are shown in Table 1. For staging, complete data—including pathology reports, imaging, and clinical examination notes—were available for 149 patients (73.8%) (Table 1).

Among all patients, the most common tumor stage was T4 (34.6%), and the most frequently observed nodal stage was N0 (36.1%). All patients were classified as M0. Overall, stage IV was the most prevalent disease stage, observed in 54.4% of cases (Appendix A).

### 3.2. Staging Results by Clinicians and Large Language Models

In assessing concordance with clinicians, ChatGPT demonstrated the highest agreement with 173 matches (85.6%), followed by Grok with 152 matches (75.2%) and DeepSeek with 136 matches (67.3%) (Table 2). The agreement in the metastasis (M) category was the highest among all of them, with a concordance rate of 100%. Significant differences were observed between ChatGPT and both DeepSeek (*p* < 0.001) and Grok (*p* = 0.009), indicating the superior performance of ChatGPT in staging accuracy. No significant difference was found between DeepSeek and Grok (*p* = 0.077) (Table 2).

Cohen’s kappa analysis revealed that ChatGPT had the highest level of agreement. All large language models demonstrated statistically significant agreement across T, N, M, and stage values according to Cohen’s kappa analysis (Figure 2, Appendix A).

The comparison of agreement and kappa statistics revealed no statistically significant difference between clinically and surgically staged patients (Appendix A). No statistically significant difference was found between the agreement rates of evaluations based on imaging, pathology, physical examination findings, and the combination of all available data (Appendix A).

Across all tumor localizations, the kappa values indicated consistent agreement for each model (Table 3).

Macro-averaged F1 scores were calculated by comparing each model’s predicted T, N, M, and overall stage values with the ground truth labels on a per-patient basis. ChatGPT achieved an F1 score of 0.78 for T classification, with DeepSeek and Grok following at 0.69 and 0.64, respectively. In N classification, scores showed greater variability; ChatGPT reached 0.86, while DeepSeek and Grok recorded 0.66 and 0.78, respectively. Overall stage prediction resulted in F1 scores of 0.85 for ChatGPT, 0.65 for DeepSeek, and 0.72 for Grok. All models correctly predicted M0 in all cases, resulting in an F1 score of 1.0 for M (Figure 3, Appendix A).

Concordance rates stratified by gender, surgical status, and TNM stage are presented in Appendix A. While no significant differences were observed according to gender or surgical status, a statistically significant variation in concordance was noted across TNM stages.

## 4. Discussion

Artificial intelligence applications are increasingly being integrated into medicine and oncology [19]. LLMs represent one of the fastest evolving and most widely adopted areas of AI by the public. To our knowledge, this is the first study to evaluate Grok and DeepSeek—two newly developed LLMs—for clinical use in oncology, specifically in the staging of HNCs.

Despite the advancements in artificial intelligence and LLM technologies, common issues such as errors and hallucinations observed across all LLMs can lead to critical consequences when used in the field of healthcare [20]. Therefore, many validation studies are currently underway to assess the reliability of LLMs in medical settings. In addition to widely used general-purpose LLMs, models specifically developed for healthcare are also being evaluated in various studies. In oncology and head and neck cancer, conformity studies with LLMs primarily focus on answering patient questions that could be directed to clinicians, answering board exam questions, and determining treatment plans [21,22,23,24,25,26,27].

Staging serves as a key predictor in both oncologic prognosis and treatment planning. Therapeutic strategies recommended by international guidelines are guided not solely by tumor resectability, but primarily by the disease stage. For example, in T1N0 nasopharyngeal carcinoma, definitive radiotherapy alone is recommended [28]. From T2N0 onward, concurrent chemoradiotherapy (CRT) becomes the standard approach. While the NCCN guideline recommends evaluating T3 disease for induction chemotherapy, the ESMO guideline explicitly supports induction chemotherapy in patients with T3N1 disease. In cases with N2 involvement, ESMO recommends adjuvant chemotherapy following CRT [29]. Likewise, T4N0 disease is also considered an indication for induction chemotherapy according to ESMO. In laryngeal cancer, the ESMO guideline recommends laser cordectomy as a treatment option for T1 tumors, whereas concurrent CRT is preferred for T1–2 tumors with N2–3 nodal disease [2]. Similarly, for oral cavity cancers, the ESMO guideline does not recommend surgery for cT4b tumors, emphasizing the importance of stage-specific non-surgical approaches in such cases [2].

There are many studies in the literature on the use of artificial intelligence applications in cancer staging. In a study leveraging pathology reports for cancer staging, the highest F1 score for T classification was achieved by the fine-tuned Clinical BigBird model (F1 = 0.81). For N classification, domain-specific clinical large language models—ClinicalCamel-70B and Med42-70B—demonstrated superior performance (F1 = 0.82–0.81) compared to the general-purpose LLaMA-2-based LLM [30].

Machine learning and large language model-based approaches have been investigated for head and neck cancer staging. In a recent study focusing on oropharyngeal cancer, the accuracy of automated T and N staging remained relatively limited (T: 55.9%, N: 56.0%), whereas M staging achieved higher performance (87.6%), in contrast to the higher staging accuracy observed in our study. In patients with lung cancer, the staging accuracy of ChatGPT using radiology and pathology reports has been reported to range between 70% and 99%, depending on the input type and the specific component of the TNM staging system [31,32]. A summary of recent studies evaluating LLMs in oncology and cancer staging is provided in Table 4.

A study comparing 50 head and neck cancer patients found that Claude provided a more accurate diagnostic approach than ChatGPT when evaluated against multidisciplinary tumor board decisions. The study, however, did not report exact staging accuracy rates [13].

Our study demonstrated that staging results incorporating surgical pathology and clinical examination notes showed similar concordance rates with radiologic/clinical staging. However, given the 85% concordance observed for ChatGPT and lower rates for other LLMs, a cautious approach is still warranted regarding the use of such models as medical assistant tools in clinical practice. While our findings suggest that LLMs may contribute to the staging process across various cancer types—particularly in head and neck cancers—prospective validation, the development of domain-specific models, clinician training, and ethical oversight will be essential to enhance their clinical applicability.

### Limitations

The limitations of this study include its retrospective design, inclusion of only non-metastatic patients, predominance of stage IV cases, and heterogeneity among the groups. The use of Turkish as the input language may have influenced model performance due to its morphologically rich structure and the limited availability of domain-specific training data [33].

While the overall distribution of disease stages reflects our real-world patient population, the relatively small number of stage I cases may have had a limited effect on the stage-specific accuracy rates. Nonetheless, this was not considered a major source of bias given the overall consistency across higher-stage groups.

A further limitation of our study is that T, N, and M components were not queried separately; instead, all results were obtained through a single-stage query. This approach may have limited our ability to assess whether stepwise prompting could improve accuracy or reduce hallucinations.

Another limitation of this study is the absence of a systematic framework for the discrete evaluation of hallucinations, out-of-scope responses, inconsistencies, and formatting errors. While such errors were implicitly captured in the overall misclassification rates and concordance analyses, they were not categorized as distinct error types. Future research should specifically address this gap by developing structured evaluation frameworks to quantify these outputs, thereby enabling a more comprehensive assessment of LLM performance in clinical oncology settings.

Finally, our study relied on a manual summarization process during case data preparation. All patient information was fully anonymized, and the summaries focused exclusively on clinical, radiological, and pathological findings relevant to TNM staging. However, no predefined structured template was used during data collection. This manual approach may carry a potential risk of human error and subjective bias. Nevertheless, this method was intentionally chosen, as in real-world clinical practice physicians typically provide LLMs with free-text case summaries rather than standardized templates.

## 5. Conclusions

This study addresses the critical need for the model-specific evaluation of large language models (LLMs) in oncology. While LLMs show promise for cancer staging, their performance varies substantially, requiring a cautious approach—particularly in heterogeneous diseases such as head and neck cancer, where staging depends on the integration of complex clinical, radiological, and pathological data. To this end, our work presents the first head-to-head comparison of three distinct LLMs—ChatGPT, DeepSeek, and Grok—using unstructured free-text records from real patients, thereby grounding the research in clinical reality. This methodology enabled a robust quantification of model performance across TNM components and subsites. By providing the first evidence of the clinical applicability of DeepSeek and Grok in oncology, our findings offer a foundational assessment of their strengths and limitations, underscoring the need for further model-specific studies to define the optimal role of each LLM in clinical practice.

## Figures and Tables

**Figure 1 diagnostics-15-02375-f001:**
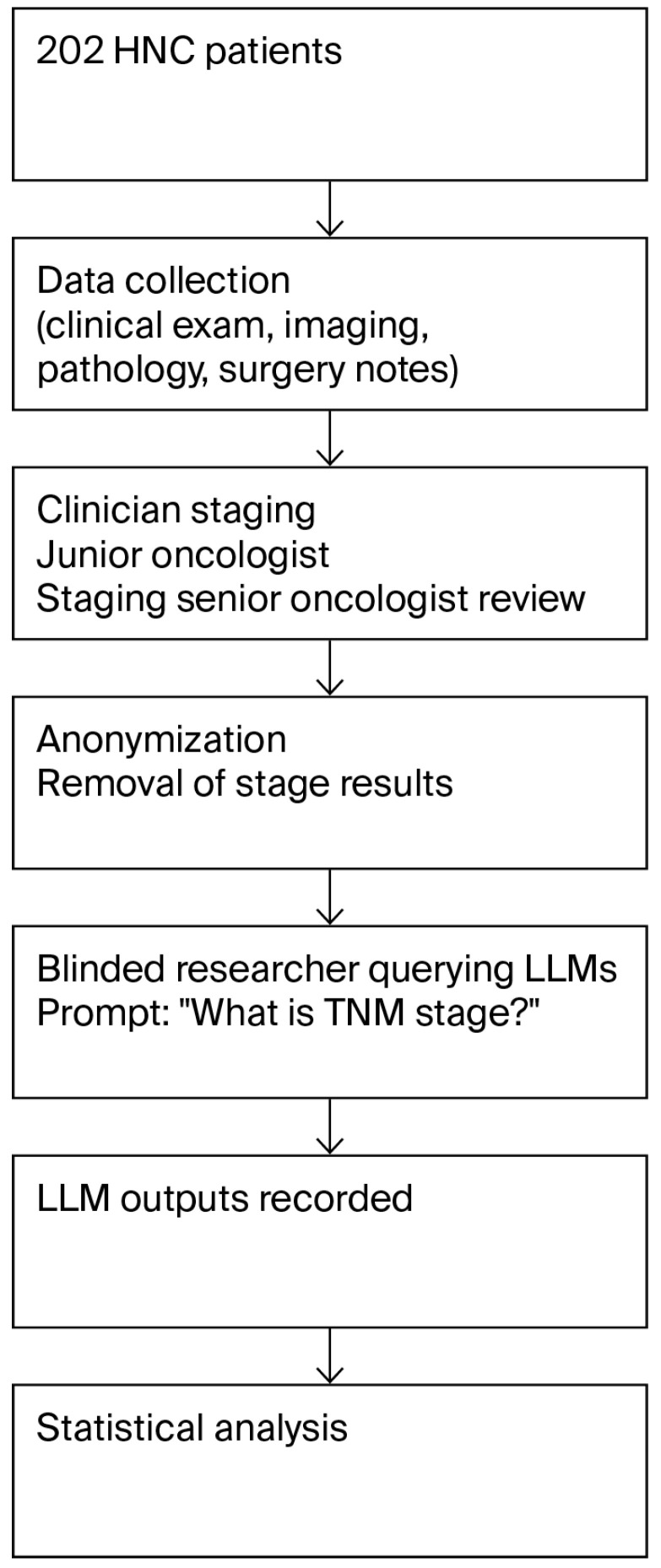
Workflow of the study.

**Figure 2 diagnostics-15-02375-f002:**
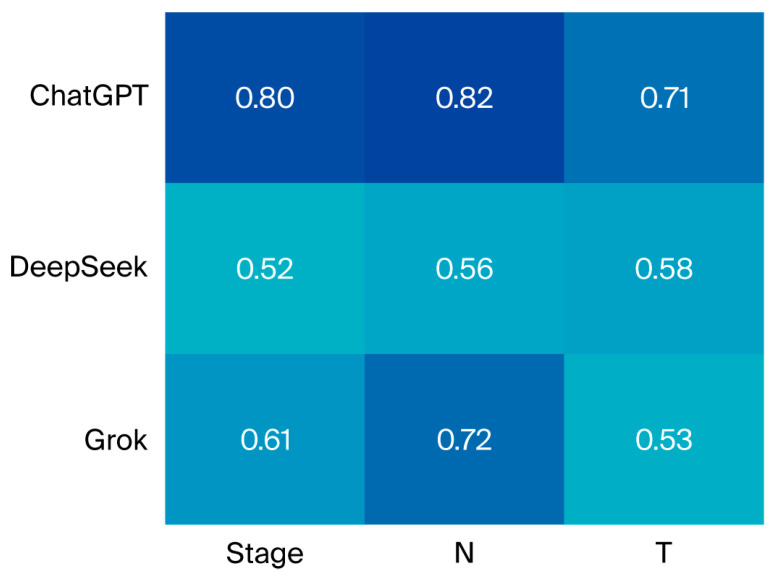
Agreement between clinicians and LLM-based models across staging systems: kappa heatmap.

**Figure 3 diagnostics-15-02375-f003:**
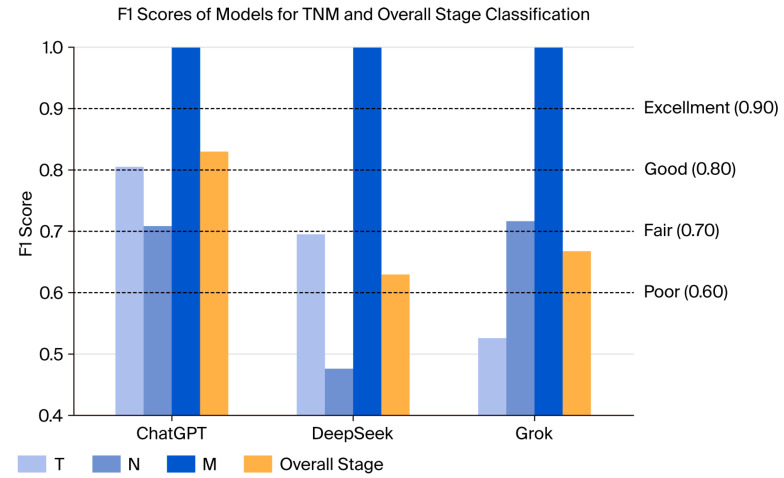
F1 scores of models for TNM and overall stage classification.

**Table 1 diagnostics-15-02375-t001:** Demographic and Clinical Characteristics of patients.

Variables	N (%)
**Gender**	
Male	151 (74.8)
Female	51 (25.2)
**Smoking**	
Active	75 (37.1)
Exsmoker	73 (36.1)
Never	51 (25.2)
**Age, years (median)**	**Median (SD)**
**Total**	58.35 (14.35)
**Male**	59.37 (13.16)
**Female**	52.86(15.67)
**Smoking Status**	
Never	54 (26.7)
Current	75 (37.1)
Ex	73 (36.1)
**Localization**	
Larynx	92 (45.5)
Hypopharynx	13 (6.4)
Oral Cavity/Oropharynx	46 (22.8)
Nasopharynx	43 (21.3)
Salivary	1 (0.5)
Nasal cavity and sinuses	7 (3.5)
**Definitive Surgery**	
Performed	80 (39.6)
Not Performed	122 (60.4)
**Diagnostic Method**	
Imaging Only	49 (24.3)
Pathology Only	3 (1.5)
Clinical Examination Note	1 (0.5)
All Methods Combined	149 (73.8)
**Stages**	
**1**	11 (5.4)
**2**	23 (11.4)
**3**	58 (28.7)
**4**	110 (54.4)

**Table 2 diagnostics-15-02375-t002:** TNM Stage Concordance Between Large Language Model and Clinician.

LLM	T Stage *n* (%)	N Stage *n* (%)	M Stage *n* (%)	TNM Stagen (%)	*p*
**ChatGPT**	159 (78.7)	176 (87.1)	202 (100)	173 (85.6)	*p*1 < 0.001 *p*2 = 0.077 *p*3 = 0.009 *p*4 < 0.001
**DeepSeek**	139 (68.8)	135 (66.8)	202(100)	136 (67.3)	
**Grok**	130 (64.4)	162 (80.2)	202 (100)	152 (75.2)	

*p*1 ChatGPTstage vs. DeepSeekstage, *p*2 DeepSeekstage vs. Grokstage, *p*3 ChatGPTstage vs. Grokstage, *p*4 collectiveS. Abbreviations: LLM = Large Language ModelBonferroni correction (α = 0.05/3 = 0.0167).

**Table 3 diagnostics-15-02375-t003:** Cohen’s kappa Values between Clinicians and LLM Staging.

LLM	Larynx *n* (%) κ (sd)	Hypopharynx *n* (%) κ (sd)	Oral Cavity/Oropharynx *n* (%) κ (sd)	Nasopharynx *n* (%) κ (sd)	Salivary *n* (%) κ (sd)	Nasal Cavity and Sinuses *n* (%) κ (sd)	Overall *n* (%) κ (sd)
**ChatGPT**	77 (83.7)	11 (84.6)	41 (89.1)	38 (88.4)	1 (100)	5 (71.4)	173 (85.6)
	0.778 (0.06)	0.743 (0.19)	0.852 (0.07)	0.791 (0.09)	-	0.553 (0.34)	0.797 (0.04)
**DeepSeek**	60 (65.2)	8 (61.5)	32 (69.6)	29 (67.4)	1 (100)	6 (85.7)	136 (67.3)
	0.518 (0.09)	0.268 (0.27)	0.580 (0.12)	0.580 (0.12)	-	0.611 (0.32)	0.522 (0.06)
**Grok**	67 (72.8)	12 (93.1)	35 (76.1)	32 (74.4)	1 (100)	5 (71.4)	152 (75.2)
	0.596 (0.08)	0.687 (0.21)	0.671 (0.11)	0.497 (0.13)	-	-	0.614 (0.05)

Abbreviations: LLM = Large language model, SD = standard deviation, k = kappa value.

**Table 4 diagnostics-15-02375-t004:** Summary of the literature on large language models (LLMs) in oncology and cancer staging.

Study (Year)	Cancer Type/Setting	LLM Evaluated	Task	Key Findings
Tozuka et al., 2025 [14]	Lung cancer (fictional cases, *n* = 100)	NotebookLM (RAG-enhanced)	TNM staging	Accuracy of ~76–86%, demonstrated feasibility of staging from reports
Chizhikova et al., 2025 [11]	Colorectal cancer	Pre-trained LLMs (BERT, RoBERTa)	Radiology report TNM extraction	Macro F1 scores of 0.7464, 0.8792, and 0.6776 for T, N, and M staging, respectively
Baran et al., 2024 [12]	Oropharyngeal cancer	AI-based NLP (health record extraction)	Automated staging	T: 55.9%, N: 56.0%, M: 87.6%, and p16: 92.1%; local vs. advanced: 80.7%
Schmidl et al., 2024 [13]	Head and neck cancer (*n* = 50)	Claude vs. ChatGPT	Diagnosis, therapy recommendations	OPSCC surgery: ChatGPT: 72.2% vs Claude: 27.8%
Lee et al., 2024 [31]	Lung cancer	ChatGPT (3 versions)	Staging using CT/PET reports	GPT-4o: 74.1%; GPT-4: 70.1%; and GPT-3.5: 57.4%

## Data Availability

The data analyzed in this study are not publicly available due to privacy and ethical restrictions involving patient information. Access to the data may be granted by the corresponding author upon reasonable request and with appropriate institutional approvals.

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
