# Peer review of "Confirmation of Large Language Models in Head and Neck Cancer Staging"

_diagnostics, 2025, doi:10.3390/diagnostics15182375_

Round 1

Reviewer 1 Report

Comments and Suggestions for Authors

The paper addresses a remarkably interesting and relevant topic, studying a contemporary issue and grounding its discussion in current literature.

Suggestions:

  1. Improve the resolution and clarity of Supplementary Figure 1 to enhance interpretability.
  2. Specify the percentage of smokers in the text to better contextualize the findings.
  3. Explain and justify the use of the authors’ native language in the study, particularly regarding its impact on data collection and interpretation, 

Author Response

Comment 1: Improve the resolution and clarity of Supplementary Figure 1 to enhance interpretability.

Response 1.

We thank the reviewer for the insightful suggestion. As requested, we have enhanced the resolution and clarity of Supplementary Figure 1 to improve interpretability. The updated figure is now provided in high resolution , with improved legibility of both the input prompt and the model response. 

Comment 2: Specify the percentage of smokers in the text to better contextualize the findings

Response 2.

We thank the reviewer for this constructive suggestion. In the revised Results section, we have now provided detailed smoking history. The text has been updated as follows: 

The majority of the patients were male (74.8%). Regarding smoking history, 37.1% were active smokers, 36.1% were former smokers, and 25.2% had never smoked

Comment 3: Explain and justify the use of the authors’ native language in the study, particularly regarding its impact on data collection and interpretation.

Response 3.

We thank the reviewer for raising this important point. As our study was conducted in Türkiye, all patient records, including outpatient clinic notes, pathology reports, and radiological assessments, were originally documented in Turkish. Using the native language ensured accurate data collection and prevented information loss or misinterpretation that might occur during translation. This approach reflects real-world clinical practice in our setting, where physicians and medical records operate in Turkish.

We also acknowledge that the use of Turkish may have influenced model performance, given its morphologically rich structure and the relatively limited availability of domain-specific training data in this language. This limitation was noted in the Discussion section of our manuscript. However, we believe that evaluating LLMs in the local language of patient care adds translational value by testing their applicability in real-world, non-English-speaking clinical environments.

Reviewer 2 Report

Comments and Suggestions for Authors

This article is impressive in terms of analysing the clinical accuracy of Large Language Models in head and neck cancer staging. However, the methods section is weak in some areas, such as the details of prompt design, systematic evaluation of hallucinations and out-of-scope responses. My recommendations for this article are given below

1) Provide a detailed analysis of query design. All prompts given to LLMs should be analysed in detail. The article should also include examples of prompts that achieved the highest success rates. Additionally, the purpose, scope/constraints, expected output format, and hallucination prevention strategies (e.g., chain-of-reasoning, ‘say if you don't know’ directive, source request) for each prompt should be clearly stated; it should be made clear whether these approaches were used.

2) Add a systematic evaluation of hallucinations and out-of-scope responses. Provide an evaluation framework not only for phase alignment but also for hallucination rate, out-of-scope information generation, inconsistencies, and format errors. Reporting these metrics will significantly simplify the performance analysis of language models.

3) Compare single requests with multiple requests and a chained approach. Many language models have been found to produce more successful results with stepwise/chained requests. Therefore, report on the performance of single requests compared to multi-step requests (e.g., first extracting TNM components separately and then calculating the stage).

4) Visualise the workflow. Add a clear flowchart showing steps such as ‘Data collection → cleaning....’ This will help the reader understand the work more easily.

5) Enrich the article with visuals. For example, include sample visuals of the data used. Also, improve the resolution of the visuals in the article.

Author Response

Comment 1) Provide a detailed analysis of query design. All prompts given to LLMs should be analysed in detail. The article should also include examples of prompts that achieved the highest success rates. Additionally, the purpose, scope/constraints, expected output format, and hallucination prevention strategies (e.g., chain-of-reasoning, ‘say if you don't know’ directive, source request) for each prompt should be clearly stated; it should be made clear whether these approaches were used.

Response 1.

We thank the reviewer for this insightful comment. As clarified in the revised Methods section, our study was intentionally designed without prompt engineering. After clinician-assigned staging was completed and entered into the database, the anonymized case information (excluding the final stage) was forwarded to a blinded third researcher. This researcher submitted the data to ChatGPT, Grok, and DeepSeek using a single standardized query: “What is the TNM stage?”.

The rationale for this design was to reflect a real-world clinical scenario, where physicians are more likely to use simple, natural queries rather than optimized or engineered prompts. By deliberately avoiding complex prompt structures, we were able to assess not only the accuracy but also the potential for hallucinated outputs under realistic usage conditions. This approach aligns with the primary aim of our study, which was to evaluate the models’ performance in an unbiased and practice-relevant manner.

Comment 2) Add a systematic evaluation of hallucinations and out-of-scope responses. Provide an evaluation framework not only for phase alignment but also for hallucination rate, out-of-scope information generation, inconsistencies, and format errors. Reporting these metrics will significantly simplify the performance analysis of language models.

Response 2.

We thank the reviewer for this valuable suggestion. In the current study, our primary focus was on concordance between clinician-assigned and LLM-generated staging results. We did not implement a separate framework to categorize hallucinations, out-of-scope responses, inconsistencies, or formatting errors. Instead, such issues were included in the overall misclassification rates reflected in our concordance, kappa, and F1 analyses.

We acknowledge this as a limitation of our study and have revised the manuscript accordingly. A note has been added to the Limitations section, stating that future studies should systematically classify and quantify hallucinations, out-of-scope information, and formatting inconsistencies to provide a more granular assessment of model performance.

Comment 3) Compare single requests with multiple requests and a chained approach. Many language models have been found to produce more successful results with stepwise/chained requests. Therefore, report on the performance of single requests compared to multi-step requests (e.g., first extracting TNM components separately and then calculating the stage).

Response 3.We thank the reviewer for this valuable suggestion. The reviewer raises an important point regarding the potential for chained prompting (e.g., extracting T, N, and M components separately before determining the final stage) to improve accuracy.

However, we deliberately employed a single-turn query for two primary reasons that were central to our study's design. First, our primary objective was to simulate a real-world clinical scenario where a clinician is most likely to issue a single, direct question rather than a multi-step command. Second, adopting a chained approach would have compromised the integrity of our blinded methodology, as the researcher entering the prompts could have inferred the final stage from the intermediate T, N, and M components. A secondary consideration was the significant increase in workload and complexity that multiple queries per case would introduce.

To address the reviewer's point, we have now clarified our rationale in the manuscript and have added the comparative performance of chained prompting as a valuable direction for future, non-blinded studies in our limitations section.

Comment 4) Visualise the workflow. Add a clear flowchart showing steps such as ‘Data collection → cleaning....’ This will help the reader understand the work more easily.

Response 4 ) 

We thank the reviewer for this helpful suggestion. In response, we have added a flowchart to the revised manuscript (Figure 1) that illustrates the overall workflow of the study, including patient selection, data collection, clinician staging, blinded LLM querying, and statistical analysis. We believe that this visual representation improves the clarity of our methodology and will help readers to more easily follow the study design.

Comment 5) Enrich the article with visuals. For example, include sample visuals of the data used. Also, improve the resolution of the visuals in the article.

We thank the reviewer for this constructive comment. In the revised version, we have enriched the article with additional visuals to enhance clarity. Specifically, we have added a sample clinical vignette and an example LLM output as supplementary figures, illustrating the type of input and responses analyzed. In addition, all figures have been regenerated in high resolution  to ensure improved readability. We believe these changes strengthen the manuscript and make it more accessible to readers.

Reviewer 3 Report

Comments and Suggestions for Authors

Dear Authors, 

Thank you for submitting the article. After the review am sharing with you that unfortunately contribution is not sufficient. Furthermore, there are some comments from the article:  

  1. Abstract needs to be rewritten in such a way that it reflects the introduction, objective, methodology, results, and conclusion.
  2. Add the literature summary table, research gaps.
  3. Properly mention the contributions of this study. 
  4. Add Reference (SPSS Version 26)
  5. Add Reference and Version (This third researcher entered the data into ChatGPT, Grok, and DeepSeek applications)
  6. Limited Contribution as mentioned in the paper (This study was designed to explore the use of ChatGPT while also evaluating DeepSeek and Grok for the first time in both oncology and head and neck cancer staging,  aiming to assess their potential in clinical applications.)
  7. Practical Implication Missing
  8. Comparison with the state-of-the-art studies is better to add. 
  9. Literature Review Summary Table with Limitations needs to be added.
Comments on the Quality of English Language

The English could be improved to more clearly express the research.

Author Response

Comment 1.Abstract needs to be rewritten in such a way that it reflects the introduction, objective, methodology, results, and conclusion.

Response 1.

Thank you for your valuable comment. We have revised the abstract to ensure that it clearly reflects the introduction, objective, methodology, results, and conclusion. The updated version has been inserted into the manuscript.

Comment 2. Add the literature summary table, research gaps.

Response 2.

Thank you for your valuable comment. In response, we have revised the manuscript by adding a comprehensive literature summary table (Table 4) that compiles prior studies evaluating LLMs in cancer staging and clinical oncology.

Comment 3. Properly mention the contributions of this study. 

Response 3.

We have revised the manuscript to better highlight the study's primary contributions. This is the first study to systematically compare ChatGPT, DeepSeek, and Grok for head and neck cancer staging. In contrast to prior research, which focused on single models or synthetic data, we utilized a real-world cohort of 202 patients with unstructured, free-text clinical, radiological, and pathological records. This approach allowed for a robust comparative analysis using concordance rates, Cohen’s kappa, and F1-scores across various anatomical subsites. Furthermore, our study introduces the first application of DeepSeek and Grok in oncology, establishing a crucial performance baseline for these newer models in a clinical setting.

Comment 4.Add Reference (SPSS Version 26)

Response 4.Thanks to reviewer.We have now added the appropriate reference for SPSS Version 26 in the Materials and Methods section.

Comment 5.Add Reference and Version (This third researcher entered the data into ChatGPT, Grok, and DeepSeek applications)

Response 5. We appreciate the reviewer's thorough review and helpful suggestions.The version of each LLM has been specified (ChatGPT Plus GPT-4o, DeepSeek V3, Grok3) and corresponding references have been added in the Methods section.

Comment 6.Limited Contribution as mentioned in the paper (This study was designed to explore the use of ChatGPT while also evaluating DeepSeek and Grok for the first time in both oncology and head and neck cancer staging,  aiming to assess their potential in clinical applications.)

Response 6. Thank you for this valuable suggestion.We clarified in both the Introduction and Discussion that this study was primarily designed to explore the use of ChatGPT, while also providing the first evaluation of DeepSeek and Grok in oncology and head and neck cancer staging, emphasizing the limited contribution and exploratory nature of the work.

Comment 7.Practical Implication Missing

Response 7 . Your valuable suggestion helped us expand the scope of our work.We expanded the Discussion to provide a more detailed comparison of our findings with state-of-the-art studies (e.g., Tozuka et al., Chizhikova et al., Baran et al., Schmidl et al., Lee et al.), showing similarities and differences in accuracy, F1 scores, and concordance.

Comment 8.Comparison with the state-of-the-art studies is better to add. 

Response 8. Thank you for this constructive feedback.In response to this comment, we included a new table (Table 4) summarizing relevant studies, their main findings, and their reported limitations. This addition provides context and clarifies where our study contributes within the existing evidence.

Comment 9.Literature Review Summary Table with Limitations needs to be added.

Response 9. Thank you for this valuable suggestion. We have added a new literature review summary table (Table 4) that compiles prior studies on LLMs in oncology and cancer staging, along with their reported limitations. This table provides a concise comparison of key findings, accuracy metrics, and methodological constraints across studies (Tozuka et al., Chizhikova et al., Baran et al., Schmidl et al., Lee et al.), alongside our current work. We believe this addition enhances the context of our findings and clearly delineates the novelty and contribution of our study.

Round 2

Reviewer 2 Report

Comments and Suggestions for Authors

The article has generally addressed my concerns and has been revised as much as possible. Therefore, I recommend its acceptance.

Reviewer 3 Report

Comments and Suggestions for Authors

Dear Authors,

Thank you to address the comments. There are some formatting issues.